# VeCAF: Vision-language Collaborative Active Finetuning with Training Objective Awareness

## ABSTRACT

Finetuning a pretrained vision model (PVM) is a common technique for learning downstream vision tasks. However, the conventional finetuning process with randomly sampled data points results in diminished training efficiency. To address this drawback, we propose a novel approach, **V**ision-*languag**e** **C**ollaborative **A**ctive **F**inetuning (`VeCAF`). With the emerging availability of labels and natural language annotations of images through web-scale crawling or controlled generation, VeCAF makes use of these information to perform parametric data selection for PVM finetuning. VeCAF incorporates the finetuning objective to select significant data points that effectively guide the PVM towards faster convergence to meet the performance goal. This process is assisted by the inherent semantic richness of the text embedding space which we use to augment image features. Furthermore, the flexibility of text-domain augmentation allows VeCAF to handle out-of-distribution scenarios without external data. Extensive experiments show the leading performance and high computational efficiency of VeCAF that is superior to baselines in both in-distribution and out-of-distribution image classification tasks. On ImageNet, VeCAF uses up to 3.3× less training batches to reach the target performance compared to full finetuning, and achieves an accuracy improvement of 2.7% over the state-of-the-art active finetuning method with the same number of batches.

## CCS CONCEPTS

• **Computing methodologies → Computer vision**.

## KEYWORDS

active learning, vision-language models, finetuning

## 1 INTRODUCTION

Deep learning has made significant progress in the field of computer vision that is typically attributed to the use of large-scale models and datasets [8, 26]. Hence, training such models from scratch is a time-consuming process and demands extensive amount of data. To address this, the pretraining-finetuning [9, 33, 39] paradigm has been recognized as a favorable approach for both vision and language tasks. For vision tasks, a model can be first trained on abundant supervised or unsupervised data and be saved as a pretrained vision model (PVM) [2, 6, 28]. Then, the PVM is finetuned on a labeled dataset for a specific downstream task. By capitalizing on ample

*ACM MM, 2024, Melbourne, Australia*
© 2024 Copyright held by the owner/author(s). Publication rights licensed to ACM.
ACM ISBN 978-x-xxxx-xxxx-x/YY/MM
https://doi.org/10.1145/nnnnnnn.nnnnnnn

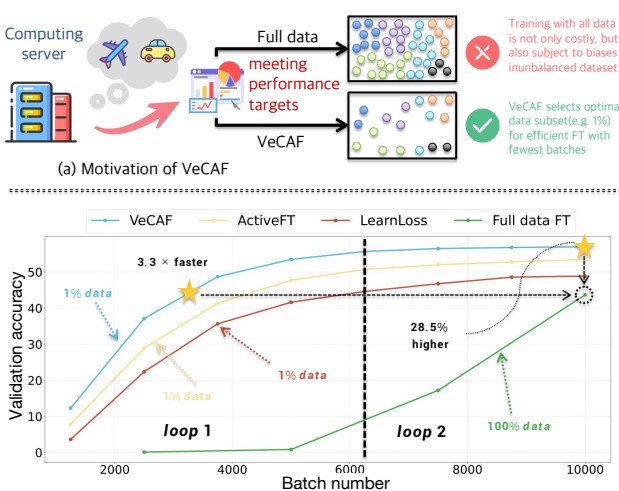

(a) Motivation of VeCAF

(b) Trade-off of training efficiency

**Figure 1: (a) Motivation of VeCAF. We select the optimal subset from a large labeled training set for efficient finetuning (FT) towards a user-specified objective. (b) Training curve comparison on ImageNet-1K validation set. All baselines select 1% of data in each FT loop with the exception of a conventional setup with full-data FT. VeCAF achieves the target accuracy faster with significantly fewer training batches and achieves higher accuracy with the same training cost.**

pretraining data and conserving valuable training resources during the finetuning stage, this paradigm has archived remarkable adoption in practical applications.

In the real-world deployments, practitioners aim to adapt deep learning models to a certain scenario or tune a model towards a specific performance target with minimal efforts for data selection and with quick training. Training with all available downstream task data can be not only costly but also can lead to a biased or degraded performance in case of improperly collected data. This motivates the proposal of a data selection framework that can actively select the optimal data subset for finetuning. Previous work on active learning [4, 41] has shown the feasibility of PVM finetuning with only a small subset (e.g., less than 5%) of training data while achieving high performance metrics in downstream tasks. However, this line of work is often limited by the setting of low label availability which hinders its effectiveness to meet the user-specified objectives.

With the growing feasibility to gather large amounts of images with labels and natural language captions in the target domain through web-scale data crawling [35] or controlled generation [25], we find it is practical to explore a novel setting of active finetuning using *annotated* data. Then, we aim to select an optimal subset of

training data for finetuning while having faster convergence and/or higher performance metrics as shown in Figure 1. The selection can further be performed in a loop to accommodate the changing model performance during the finetuning process. To this end, we propose to perform an Objective-aware Data Selection (ODS) using a parameterized data selection model. This ODS model reweighs training data distribution according to the downstream objective and selects a subset that is both diverse and representative to the task.

The pursuit of objective-awareness brings new challenges to the data selection. Intuitively, images with misleading object appearances and complicated backgrounds, as illustrated in Figure 2, often provide more informative supervision. However, the image features extracted by the PVM may not fully capture all the semantic information present in the image. Therefore, PVM image features may miss useful information for the data selection and finetuning process as in previous ActiveFT [41] work. To address this limitation, we propose to leverage semantically rich language embedding spaces of the text encoders (e.g., CLIP [33], mT5 [42], BERT [9] etc.) in our novel **V**ision-languag**e C**ollaborative **A**ctive **F**ine-tuning (VeCAF) approach. Specifically, we extract the text embeddings of the captions associated with each image. These captions may be sourced directly from original datasets such as COCO-Caption or alternatively generated by a multimodal Large Language Model (LLM) e.g., BLIP-2 [22]. Then, we propose a Cross-attentive Embedding Augmentation (CEA) to augment the image features extracted by the PVM such that the augmented features can focus more on the rich semantic information of the training samples. Therefore, the CEA facilitates both the active data selection and the finetuning process.

Empirically, we demonstrate improved efficiency and performance across different scenarios. VeCAF is evaluated on three image classification datasets including CIFAR-10 [19], Caltech101 [12], and ImageNet-1K [8] with the pretraining-finetuning paradigm where base models are pretrained on ImageNet-1K. Results on ImageNet demonstrate that VeCAF can significantly accelerate the PVM convergence speed to the target performance and saves up to 3.3× computational cost when compared to finetuning with all training set. Importantly, VeCAF also addresses out-of-distribution (OOD) scenarios, where we augment image features by target-domain text embeddings derived from the generated image captions. By leveraging the alignment between text and images embeddings, VeCAF increases the likelihood of selecting images that possess the characteristic features of the target domain from the training dataset as shown in Figure 3. In addition, we verify the OOD generalization ability on the corrupted ImageNet-C dataset where VeCAF improves accuracy by over 6% when compared to state-of-the-art active learning methods. Our main contributions are summarized as follows:

- We propose a novel framework, VeCAF, to improve computational efficiency of PVM finetuning using both the training objective and the language-embedded knowledge.
- We propose the Objective-aware Data Selection (ODS), where a parameterized data selection model is optimized for the user-specified objectives and selects a subset that contributes to faster convergence and higher performance metrics.
- We further employ pretrained language encoders with the proposed Cross-attentive Embedding Augmentation (CEA) to

enrich semantic information in image features and to provide explicit semantic guidance for data selection and finetuning.

## 2 RELATED WORK

### 2.1 Active Learning

The learning algorithm in active learning is allowed to choose the data from which it learns. There are two main selection criteria: uncertainty [3, 27, 44, 46] and diversity [1, 5, 40]. Uncertainty of the model can aid selection of the most difficult unlabeled data. Early works estimate the uncertainty with various heuristics such as posterior probability [21, 45], entropy [17, 29] and classification margin [37]. Previous works [13, 31, 36] also formulate active learning as an optimization problem. They typically operate in a discrete space that trivially matches the sample distribution of a dataset [10, 15]. However, discrete optimization is harder to solve than the one in continuous space. Also, most previous methods are designed for a from-scratch training without the pretraining stage. Bengar et al. [4] reveals drawbacks of such setting without unsupervised pretraining. Xie et al. [41] addresses both shortcomings with the proposed continuous-space ActiveFT method that applies selected samples to the finetuning of the pretrained model in a single pass. VeCAF extends ActiveFT to the practical setting of data selection from a large labeled dataset. VeCAF framework enhances the training efficiency with a training objective-aware data selection and achieves optimal finetuning results with minimal training batches.

### 2.2 Exploiting Language in Vision Model Training

Recent developments in language models, especially vision-language models [22, 23] demonstrate their effectiveness in aligning the embedding space of vision and language to achieve cross-modal generalization. For example, BLIP-2 [22] addresses the modality gap using a lightweight Querying Transformer, while Shikra [7] handles spatial coordinate inputs and outputs in natural language and excels in referential dialogue and general vision-language tasks. Many prior work exploit the connection between the image and text modalities, where they explore the use of language in training better vision models. For example, Ma et al. [30] leverage pretrained language models to design a distribution alignment objective. This objective guides the vision model to learn linguistic representations specific to the task under a semi-supervised setting. Similarly, Fahes et al. [11] utilize CLIP to optimize affine transformations of source domain features. This optimization aligns these features with the target text embeddings while preserving their content and semantics. In our work, we use the language embeddings of image captions to perform text-space augmentations, achieving better sample selection quality in active learning setting.

## 3 PROPOSED METHOD

This section introduces the details of the proposed VeCAF framework. As illustrated in Figure 2, we start by selecting a subset of training data with the PVM image feature space using the proposed ODS method, as introduced in Section 3.1. Then, the selected samples pass through pretrained language model to get semantically-rich text embeddings, which augment the image features with our proposed CEA technique, as formulated in Section 3.2. The augmented

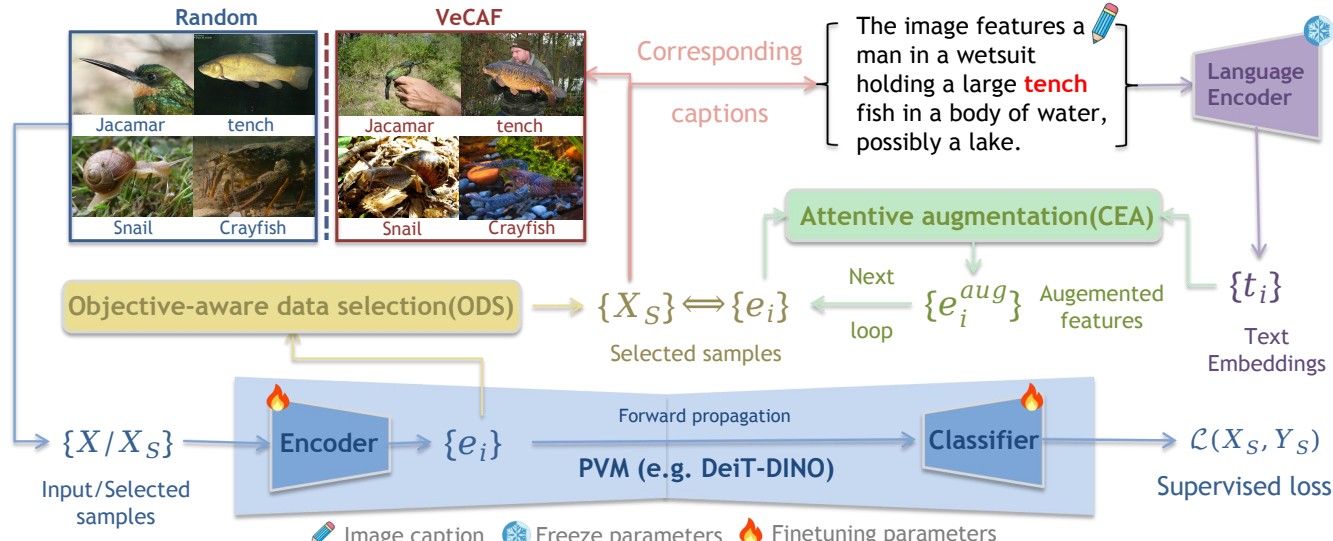

**Figure 2: The overall framework of VeCAF. In each data selection loop, VeCAF performs an Objective-aware Data Selection (ODS) to select more informative images for finetuning. Cross-attentive Embedding Augmentation (CEA) is performed on the selected images to further enrich the semantic information captured by the image embeddings by incorporating language knowledge of the caption.**

image features are used for PVM finetuning as candidates in future rounds of data selection. Finally, we show that VeCAF can overcome challenges of out-of-distribution data in Section 3.3.

## 3.1 Training Objective-Aware Data Selection

Consider a PVM $f(\cdot; w)$ with $w$ weights and a user-defined finetuning objective $\mathcal{L}$. Given a labeled training set $\mathcal{D} : \{\mathcal{X}, \mathcal{Y}\}$, where $\mathcal{X}$ is the set of image $x$ and $\mathcal{Y}$ is the set of corresponding label $y$. The goal of our active data selection is to find a subset $\mathcal{S} : \{\mathcal{X}_S, \mathcal{Y}_S\} \subset \mathcal{D}$, such that PVM finetuning with this subset for a fixed number of iterations leads to the largest reduction of the training objective. Formally, we formulate the optimization problem of our objective-aware data selection (ODS) process as

$$\mathcal{S}_{opt} = \arg\min_{\mathcal{S}} \mathbb{E}_{x,y \in \mathcal{D}} \left[ \mathcal{L}\left(f(x; w - \beta \delta_S), y\right) \right], \quad (1)$$

where $\delta_S = \nabla_w \mathbb{E}_{x_s, y_s \in \mathcal{S}} \left[ \mathcal{L}\left(f(x_s; w), y_s\right) \right]$ is the gradient accumulated by finetuning with $\mathcal{S}$ and $\beta$ is the learning rate.

In a practical setting, we cannot compute $\nabla_w \mathcal{L}(f(x; w), y)$ for each training example $(x, y)$ before the data selection, as the gradient computation cost almost equals to the cost of full finetuning, which contradicts the purpose of active data selection. In this sense, we make an assumption that a data point with a larger loss contributes more to the convergence speed of the model.

Intuitively, this assumption leads a naive data selection policy of selecting the data points with Top-$K$ training losses. However, previous active learning work [41] has discovered that the diversity of the selected data is also important in order to cover corner cases in the dataset and to avoid overfitting. Therefore, we design the ODS algorithm with the following principles: **1)** *data point with a larger loss $\mathcal{L}(f(x; w), y)$ shall be selected with a higher probability*; and **2)** *maintaining the diversity of data points selected in $\mathcal{S}$*. Analytically,

we formulate our ODS objective as

$$\mathcal{S}_{opt} = \arg\min_{\mathcal{S}} D_{KL}(p_{\mathcal{L}}(\mathcal{D})||p_S(\mathcal{S})) - \lambda R(p_S(\mathcal{S})), \quad (2)$$

where $D_{KL}(\cdot||\cdot)$ is the KL divergence, $R(\cdot)$ is a diversity metric, and $\lambda$ is a tradeoff factor. $p_{\mathcal{L}}(\mathcal{D})$ is the distribution of the full training set that guides data selection. To follow our first principle, we assign the probability of each data point $(x, y)$ in $p_{\mathcal{L}}(\mathcal{D})$ according to the finetuning objective $\mathcal{L}$ scaled by a $Z$ normalization factor as

$$p_{\mathcal{L}}(x, y) = \mathcal{L}\left(f(x; w), y\right) / Z. \quad (3)$$

The distribution of the selected data $p_S(\mathcal{S})$ is determined by the data selection model. As a sanity check, the naive "Top-$K$ training losses" serves as the optimal solution for Equation 2 without considering diversity when $\lambda = 0$.

To enable continuous optimization, we optimize Equation 2 using a parameterized data selection model $\theta_S$. For the simplicity, we follow previous work [41] to model the data selection distribution $p_S(\mathcal{S})$ in the lower-dimension image embedding space, where embedding $e$ is produced by the PVM from the input $x$ at a hidden layer. $\theta_S$ consists of $K$ centroids in the image embedding space, each selecting the nearest data point. We define the probability of a data point $x_i$ with the corresponding embedding $e_i$ being selected as

$$p_S(x_i) = \exp(\langle e_i, \theta_S^{c_i} \rangle) / \sum_{x_j \in \mathcal{D}} \exp(\langle e_j, \theta_S^{c_j} \rangle), \quad (4)$$

where $\langle \cdot, \cdot \rangle$ denotes the cosine distance, and $\theta_S^{c_i}$ is the closest centroid in $\theta_S$ to $e_i$. We derive the parameterized distribution distance $D(\theta_S) := D_{KL}(p_{\mathcal{L}}||p_S)$ for optimization objective Equation 2 using Equation 3 and 4 as

$$D(\theta_S) = \sum_{(x_i, y_i) \in \mathcal{D}} p_{\mathcal{L}}(x_i, y_i) \log \frac{p_{\mathcal{L}}(x_i, y_i)}{p_S(x_i)}$$
$$= \mathbb{E}_{p_{\mathcal{L}}}[\log p_{\mathcal{L}}(x_i, y_i)] - \mathbb{E}_{p_{\mathcal{L}}}[\log p_S(x_i)] \quad (5)$$
$$= C - \alpha \sum_{(x_i, y_i) \in \mathcal{D}} \mathcal{L}\left(f(x_i; w), y_i\right) \langle e_i, \theta_S^{c_i} \rangle,$$

where $C$ and $\alpha$ are the constants omitted in the formulation.

For the diversity metric $R(\cdot)$, we follow the diversity regularization term proposed in [41] as

$$R(\theta_S) = - \sum_{\theta_S^i} \left[ \log \sum_{\theta_S^j, j \neq i} \exp(\langle \theta_S^i, \theta_S^j \rangle) \right]. \quad (6)$$

By substituting Equation 5 and Equation 6 into Equation 2 and by removing constant terms, our final objective in terms of $\theta_S$ parameters can be written as

$$\theta_S^* = \arg\min_{\theta_S} - \sum_{(x_i, y_i) \in \mathcal{D}} \mathcal{L}\left(f(x_i; w), y_i\right) \langle e_i, \theta_S^{c_i} \rangle$$
$$+ \lambda \sum_{\theta_S^i} \left[ \log \sum_{\theta_S^j, j \neq i} \exp(\langle \theta_S^i, \theta_S^j \rangle) \right]. \quad (7)$$

The optimization on $\theta_S$ is conducted via gradient descent. To further resolve the dependency of the optimized data selection model to its initialization as observed in [41], we consider an independently-initialized data selection model ensemble[], denoted as $\{\theta_e\}_{e=1}^E$, in the optimization. We empirically set $E = 5$ to balance the performance-cost tradeoff. After optimizing each data selection model $\theta_e$ independently using Equation 7, we can then remove the bias in the initialization by utilizing the mean $\mu$ and covariance $\Sigma$ calculated on $\theta_e^*$. Specifically, given an optimized data selection model $\theta_1^*$, we achieve the final unbiased data selection model as $\theta_S^* = \theta_1^* - \Sigma^{-1}(\theta_1^* - \mu)$.

The optimization in Equation (7) is performed before each finetuning "loop" with the current model weights $w$. The training data with the closest embedding to each $\theta_S$ centroid is selected to form a "finetuning set" with $K$ elements. Then, this set is used to finetune the PVM until the next loop starts. We update ODS weights $w$ after a predetermined number of training batches.

## 3.2 Cross-Attentive Embedding Augmentation

We empirically find that samples selected by the ODS process tend to consist of multiple objects in both foreground and background, which is a result of coarse image embeddings produced by the PVM. To further improve quality of image embeddings for both data selection and finetuning, we propose Cross-attentive Embedding Augmentation (CEA). Given a training image, CEA leverages a pretrained text encoder transfer the image caption to the corresponding text embedding. Then, the text embedding is used to augment the image embedding with attention-based method.

Given selected sample $\{x_{S_i}\}_{i=1}^K$ in the previous ODS run, the corresponding image caption can be denoted as $\{c_{S_i}\}_{i=1}^K$. Then we feed the caption into a frozen text encoder, e.g. BERT [9], to convert the captions into the text embeddings $\{t_i\}_{i=1}^K$. We use text embeddings with the same dimensions as image embeddings for easier fusion.

Inspired by [16], CEA is conducted as mapping the image embedding $e_i$ towards the corresponding text embedding $t_i$. To decide the

---

**Algorithm 1:** Vision-language Collaborative Active Finetuning (VeCAF)

**input :** Objective-aware data selection $ODS(\cdot; \theta)$, labeled data pool $(\mathcal{X}, \mathcal{Y})$, image caption pool $Cap$, PVM $f(\cdot; w)$, pretrained language encoder $LM(\cdot)$, data selection loop number $L$, batch number $B$ for each loop

**output :** Finetuned vision model $f(\cdot; w_{FT})$

1 **for** $loop \in [L]$ **do**
2     Obtain data selection model $\theta_s^*$ for $f(\cdot; w)$ with Equ. (7) ;
3     Obtained the selected sample pool $\mathcal{S}_{opt} = ODS((\mathcal{X}, \mathcal{Y}); \theta_s^*)$ ;
4     Get the corresponding image caption $Cap_i$ for $s_i \in \mathcal{S}_{opt}$ ;
5     Transfer the image caption to text embedding as $t_i = LM(Cap_i)$ ;
    /* Cross-attentive embedding augmentation (CEA) */
6     CEA attention score computation $\alpha_i = \text{Softmax}(\frac{e_j \cdot t_j}{\|e_j\|_2 \|t_j\|_2})$ ;
7     Image embedding $e_i$ augmentation $e_i^{aug} = e_i - \eta \cdot \alpha_i(e_i - t_i)$ ;
    /* PVM finetuning with $\mathcal{S}_{opt}$ */
8     **for** $batch \in [B]$ **do**
9        Sample next batch from $\mathcal{S}_{opt}$ ;
10        Calculate the loss with the classifier ;
11        Optimize $f(\cdot; w_{FT})$ via gradient descent ;
12     **end**
13     $f(\cdot; w) \leftarrow f(\cdot; w_{FT})$
14 **end**
15 Return the finetuned vision model $f(\cdot; w_{FT})$

---

magnitude of the augmentation, we compute a sample-wise attention score $\{\alpha_i\}_{i=1}^K$ with the cosine distance between $e_i$ and $t_i$ as

$$\alpha_i = \text{Softmax}(\frac{e_j \cdot t_j}{\|e_j\|_2 \|t_j\|_2}) = \frac{\exp\left(\frac{e_i \cdot t_i}{\|e_i\|_2 \|t_i\|_2}\right)}{\sum_{j=1}^K \exp\left(\frac{e_j \cdot t_j}{\|e_j\|_2 \|t_j\|_2}\right)}. \quad (8)$$

The attention score $\alpha_i$ helps to derive the augmented embedding $e_i^{aug}$ using the image embedding $e_i$ and the corresponding text embedding $t_i$ as

$$e_i^{aug} = e_i - \eta\alpha_i(e_i - t_i), \quad (9)$$

where $\eta$ is the fixed step size. The proposed CEA method enriches the semantic information and improves performance after finetuning as shown in experiments. We further present the pseudo code of VeCAF in Algorithm 1, which specifies the complete procedure of the proposed Vision-language Collaborative Active Finetuning (VeCAF) framework.

## 3.3 Improving Out-of-Distribution Scenarios

In addition to achieving more efficient finetuning on in-distribution (ID) tasks, one of the additional benefits of introducing the text modality is the ability to artificially modify the corresponding image captions as a semantic augmentation. By leveraging the strong language capability provided by the text encoder, we can implicitly alter

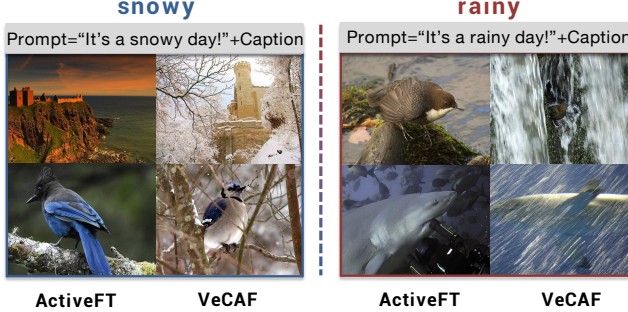

**snowy**       **rainy**

Prompt="It's a snowy day!"+Caption    Prompt="It's a rainy day!"+Caption

**ActiveFT**    **VeCAF**      **ActiveFT**    **VeCAF**

**Figure 3: The selected samples of ActiveFT [41] and VeCAF. With the caption augmented: "*It is a* {*snowy/rainy*} *day!*", Ve-CAF can select images that correspond to the target domain.**

our requirements for the selected data. This capability enables us to facilitate domain transfer to out-of-distribution (OOD) scenarios with only ID data.

For example, leveraging the CEA technique, we can add descriptive phrases like "*It's a {target_domain} day!*" to the image captions. This modification shifts the PVM image embedding towards the "snow" distribution while highlighting the ID sample images that have snowflakes or similar patterns during the data selection process, as demonstrated in Figure 3. By incorporating such textual cues, we can guide the selection process toward images that possess specific characteristics or attributes, so the finetuned model can better generalize in out-of-distribution scenarios.

## 4 EXPERIMENTS

In this section, we present results of conducted experiments on multiple image classification tasks. First, we introduce the experiment setup and evaluation protocols in Sec. 4.1. Main results in Sec. 4.2 show the effectiveness and efficiency of the proposed VeCAF when compared to other sample selection methods. Particularly, we demonstrate advantages of the proposed VeCAF in both in-distribution and out-of-distribution scenarios where it improves the finetuning efficiency by up to 3.3× and 2.7% accuracy on ImageNet-1K. In Sec. 4.3, we present further analysis including ablation studies, VeCAF generality and qualitative results.

### 4.1 Experiment Setup

*Datasets.* For model training, we conduct experiments using three image classification datasets: CIFAR-10 [19], class-unbalanced Caltech101 [12], and ImageNet-1K [8]. Details of the dataset can be found in Appendix A. For out-of-distribution evaluation, we evaluate on ImageNet-C [14] which consists of various synthetically generated corruptions applied to the ImageNet validation set.

*Implementation details.* In our main experiments, we use the DeiT-B model [38] pretrained with DINO [6] on ImageNet-1K [8] as the PVM for finetuning. Additionally, we present experiments for different PVM architectures and model sizes in Section 4.3 to demonstrate VeCAF generalizability. For all experiments, we resize input images to 224×224 to ensure consistency during both the data selection and the finetuning. In the ODS process, we optimize the

data selection model parameters $\theta$ using the Adam optimizer with a learning rate of 0.001 until convergence.

We adopt the standard protocols outlined in [38] to finetune the DeiT-B model. We use the SGD optimizer for supervised finetuning with the following hyperparameters: 5e-4 learning rate (lr), 1e-4 weight decay and 0.9 momentum. The total number of finetuning batches for CIFAR-10, Caltech101 and ImageNet is set to be 750, 1500 and 12500, respectively. The total batch numbers are set such that baseline active learning methods can converge to a reasonable performance on each dataset. All experiments are conducted on two Tesla-A100 GPUs with a batch size of 256. We finetune the model following the settings provided in the official DeiT code[1], with cosine learning rate decay applied.

*Evaluation protocol.* We primarily focus on an efficient training, where all baselines are allowed to have the same batch size and the same number of batches. Convergence results with unlimited batches are in Appendix C.3. We use a multi-run data selection as the default setting for VeCAF and all other baselines. Multi-run selection is the setting where we select a set of training data at the beginning of each selection loop and use this set for PVM finetuning during the whole loop. Having multiple loops of selection helps active learning methods as it allows the data selection model to follow the training dynamic of the PVM more closely. To balance the benefit of multi-run selection with an additional overhead of running the data selection model, we empirically set to perform 3 data selection loops in all the main experiments. Each loop divides the total number of training batches uniformly and, hence, contains 1/3-rd of the total number of training batches. This setting enables the overhead of data selection to be negligible compared to the overall training time in each data selection loop, as analyzed in Appendix C.2.

*Baselines.* We compare VeCAF with three active learning baselines LearnLoss [43], TA-VAAL [18], ALFA-Mix [32], ActiveFT [41] and the conventional approach of full data finetuning. The details of the baselines are summarized as follows:

- LearnLoss [43] predicts target losses for unlabeled inputs to select data points with potential incorrect predictions.
- TA-VAAL [18] selects data based on an estimation of the data distribution of labeled and unlabeled pools, which is further enhances by incorporating a loss prediction ranking.
- ALFA-Mix [32] employs interpolations between labeled and unlabeled instances to uncover unrecognized features, which leads to an efficient data selection policy with a closed-form solution to identify the optimal interpolation that induces changes in predictions.
- ActiveFT [41] performs data selection in the embedding space of a pretrained model, where the selection is converted to a continuous optimization to match the distribution of the data pool while maintaining diversity within selected subsets.
- Full Data finetuning (FT) randomly samples all data points from the training set to perform finetuning of the pretrained vision models following the conventional paradigm.

---

[1] https://github.com/facebookresearch/deit

**Table 1: Classification accuracy with the fixed training cost: all methods for each dataset are trained using the fixed number of batches. The percentage value on top reports the ratio of data selected during each loop. Some results marked as N/A ("-") as explained in Appendix B. Top-1 accuracy with a standard error of 3 repetitions is reported, %.**

| Method | Loop | CIFAR-10 | | | Caltech101 | | | ImageNet-1K | | |
|--------|------|------|------|------|------|------|------|------|------|------|
| | | 1% | 2% | 5% | 2% | 5% | 10% | 1% | 2% | 4% |
| LearnLoss [43] | single-run | $85.93_{\pm0.05}$ | $91.22_{\pm0.08}$ | $93.89_{\pm0.07}$ | $46.47_{\pm0.16}$ | $43.74_{\pm0.13}$ | $65.59_{\pm0.05}$ | $49.37_{\pm0.09}$ | $57.86_{\pm0.07}$ | $63.46_{\pm0.11}$ |
| | multi-run | $87.53_{\pm0.14}$ | $92.53_{\pm0.17}$ | $94.43_{\pm0.22}$ | $47.36_{\pm0.13}$ | $44.27_{\pm0.19}$ | $66.27_{\pm0.14}$ | $49.89_{\pm0.16}$ | $58.22_{\pm0.17}$ | $64.18_{\pm0.13}$ |
| TA-VAAL [18] | single-run | $85.46_{\pm0.10}$ | $92.65_{\pm0.06}$ | $94.85_{\pm0.10}$ | $59.26_{\pm0.05}$ | $58.11_{\pm0.08}$ | $66.94_{\pm0.08}$ | - | - | $63.86_{\pm0.13}$ |
| | multi-run | $87.74_{\pm0.17}$ | $93.77_{\pm0.19}$ | $96.01_{\pm0.12}$ | $60.57_{\pm0.18}$ | $59.25_{\pm0.15}$ | $67.32_{\pm0.21}$ | - | - | $64.32_{\pm0.16}$ |
| ALFA-Mix [32] | single-run | $86.69_{\pm0.07}$ | $92.87_{\pm0.06}$ | $95.14_{\pm0.08}$ | $59.73_{\pm0.05}$ | $58.74_{\pm0.09}$ | $67.36_{\pm0.08}$ | - | - | $64.03_{\pm0.07}$ |
| | multi-run | $88.14_{\pm0.13}$ | $93.26_{\pm0.13}$ | $95.75_{\pm0.11}$ | $60.74_{\pm0.16}$ | $60.47_{\pm0.14}$ | $68.25_{\pm0.15}$ | - | - | $64.69_{\pm0.19}$ |
| ActiveFT [41] | single-run | $90.91_{\pm0.12}$ | $93.80_{\pm0.09}$ | $95.39_{\pm0.08}$ | $62.86_{\pm0.05}$ | $60.55_{\pm0.09}$ | $69.34_{\pm0.06}$ | $53.96_{\pm0.07}$ | $60.33_{\pm0.09}$ | $64.72_{\pm0.10}$ |
| | multi-run | $92.79_{\pm0.11}$ | $94.17_{\pm0.16}$ | $95.92_{\pm0.14}$ | $64.43_{\pm0.12}$ | $61.97_{\pm0.17}$ | $71.42_{\pm0.14}$ | $55.67_{\pm0.13}$ | $61.86_{\pm0.21}$ | $65.18_{\pm0.17}$ |
| Full Data FT | single-run | | $93.64_{\pm0.02}$ | | | $62.08_{\pm0.02}$ | | | $57.53_{\pm0.01}$ | |
| VeCAF (ours) | multi-run | $\mathbf{93.57}_{\pm0.02}$ | $\mathbf{95.27}_{\pm0.04}$ | $\mathbf{96.24}_{\pm0.02}$ | $\mathbf{66.33}_{\pm0.04}$ | $\mathbf{65.15}_{\pm0.03}$ | $\mathbf{72.21}_{\pm0.03}$ | $\mathbf{58.31}_{\pm0.04}$ | $\mathbf{63.76}_{\pm0.03}$ | $\mathbf{66.57}_{\pm0.02}$ |

**Table 2: 1-vs.-all accuracy for certain categories on CIFAR-10. DINO-S [18] is used as PVM with 2% of data in each loop.**

| Methods | Airplane | Bird | Cat | Deer |
|---------|----------|------|-----|------|
| ActiveFT [41] | $89.50_{\pm0.01}$ | $82.70_{\pm0.03}$ | $83.00_{\pm0.02}$ | $87.50_{\pm0.04}$ |
| Top-$K$ Loss | $82.18_{\pm0.05}$ | $81.88_{\pm0.01}$ | $75.72_{\pm0.04}$ | $84.64_{\pm0.03}$ |
| VeCAF (ours) | $\mathbf{91.24}_{\pm0.02}$ | $\mathbf{88.80}_{\pm0.04}$ | $\mathbf{86.41}_{\pm0.02}$ | $\mathbf{89.12}_{\pm0.03}$ |

## 4.2 Main Results

*In-distribution performance.* We present the image classification results in Table 1. The results demonstrate the effectiveness of the proposed VeCAF method when compared to other active learning approaches. Under the fair comparison of the multi-run setting, VeCAF consistently outperforms other methods across all three datasets, irrespective of the employed sampling ratios. Remarkably, even with low data sampling ratios, our method excels in selecting highly representative samples. When compared to full data finetuning with limited training batches, VeCAF achieves higher accuracy even with only 1% of the data being used for finetuning in each epoch. Additionally, it is noteworthy that the performance gain brought by VeCAF over previous active learning methods increases with the training set being more complex. For example, VeCAF gains 2.7% accuracy over ActiveFT on ImageNet-1K compared to less than 1% on CIFAR-10, which proves that the VeCAF is suitable for more complex learning tasks. In summary, these results demonstrate VeCAF to be more effective with the pretraining-finetuning paradigm when compared to previous active learning methods.

*Finegrained objective-aware training.* The proposed ODS method offers additional flexibility to accommodate finegrained training objectives. For verification, we evaluate VeCAF under 1-vs.-all finetuning objective on multiple CIFAR-10 classes. Specifically, given a target class, we set the loss as a binary classification, with target

**Table 3: OOD generalization ability. Models are trained with 1% data subset per loop using the uncorrupted ImageNet. Evaluation results are for the distorted ImageNet-C validation set.**

| Source | Target (eval.) | Method | Top-1 Acc. |
|--------|----------------|--------|------------|
| ImageNet | | Prompt: It's a snowy day. | |
| | ImageNet-C Snowy | Full Data FT | $38.89_{\pm0.20}$ |
| | | CLIPStyler | $40.71_{\pm0.41}$ |
| | | ActiveFT | $36.36_{\pm0.11}$ |
| | | VeCAF | $\mathbf{42.33}_{\pm0.03}$ |
| | | Prompt: It's a foggy day. | |
| | ImageNet-C Foggy | Full Data FT | $45.55_{\pm0.27}$ |
| | | CLIPStyler | $46.25_{\pm0.36}$ |
| | | ActiveFT | $42.45_{\pm0.08}$ |
| | | VeCAF | $\mathbf{47.71}_{\pm0.03}$ |

class being positive and all others being negative. The modified loss is used in ODS optimization as in Equation (7). A naive objective-aware baseline of selecting samples with the largest loss is also included in the comparison. Results in Table 2 show that VeCAF performs better with the finegrained objectives for target categories, and the naive addition of top-$K$ loss is the inadequate approach for data selection with finegrained objectives.

*Out-of-distribution generalization.* We further conduct experiments using ImageNet-C validation set to verify the capability of VeCAF on helping a model finetuned only on the source domain to generalize towards out-of-distribution (OOD) scenarios. Specifically, we consider the domain adaptation scenarios of clear→snowy and clear→foggy. The results are presented in Table 3.

In these experiments, we finetune a PVM classifier using only uncorrupted (source) ImageNet data points. The only exception is CLIPstyler [20] which is the state-of-the-art domain transfer framework. It generates stylized images of the target domain from source

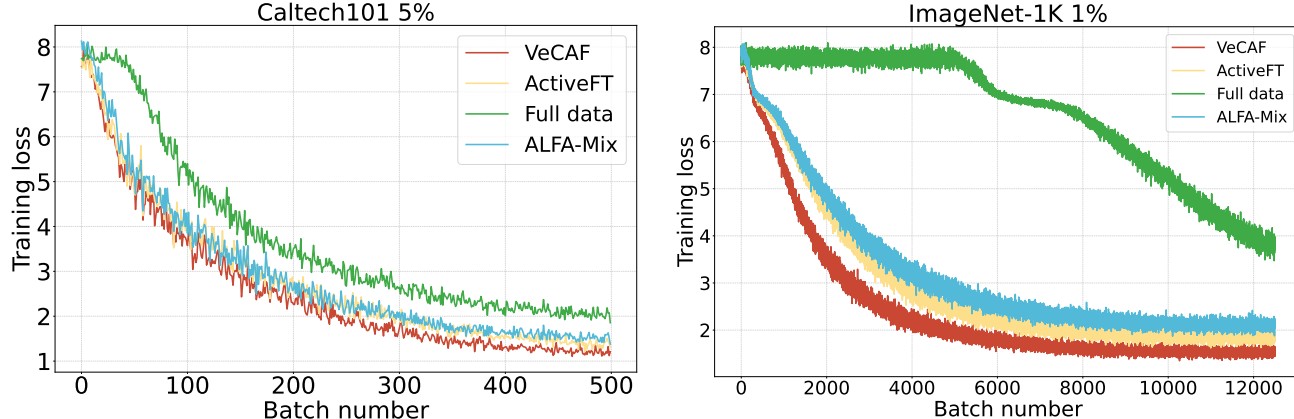

**Figure 4: Training loss curve of VeCAF and other baselines including ActiveFT, ALFA-Mix, and Full data FT on Caltech-101 (left) and ImageNet-1K (right) with 5% and 1% data, respectively.**

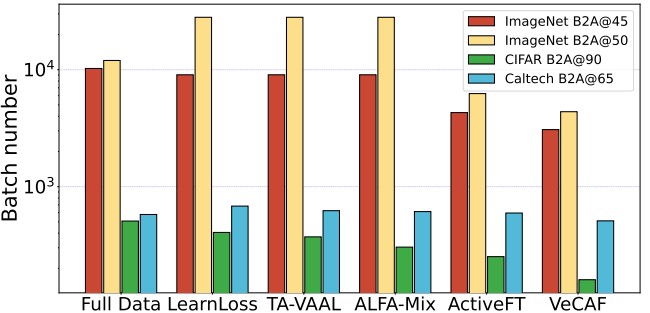

**Figure 5: Comparison of training efficiency. VeCAF requires significantly fewer training batches to reach the target accuracy (B2A) compared with other baselines and full-data finetuning. Note that the y-axis has an exponential scale.**

domain images by leveraging the knowledge of pretrained CLIP models. Finetuning settings are the same in the previous experiments on ImageNet with 1% of the training data being used in each finetuning loop for ActiveFT and VeCAF. We can see from Table 3 that VeCAF consistently outperforms three baselines by up to 4.2%. CLIPstyler helps OOD generalization when compared to Full Data FT, but the presence of artifacts in the stylized images limits its performance. Despite that ActiveFT has good performance on in-distribution data, it underperforms in the OOD setting. This is a result of overfitting because ActiveFT selects data only based on the source domain distribution. On the other hand, the proposed domain-specific text embedding augmentation in VeCAF allows the source domain data that resembles the target domain style to be selected in the finetuning process (see Figure 3). This leads to better generalization in OOD scenarios.

*Efficiency analysis.* To provide a deeper understanding of the potential training efficiency improvement introduced by VeCAF, we illustrate the required batch numbers to achieve the target accuracy (B2A) for different approaches on the three datasets. Specifically, we report training batches needed for each method with **B2A@45,50**

**Table 4: FLOPs analysis for VeCAF and Full-FT**

| Method | Training batches | Total FLOPs (G, all batches) | | | |
| --- | --- | --- | --- | --- | --- |
| | | CEA | ODS/DS | FT | ALL |
| VeCAF | $3 \times 10^3$ | $9.53 \times 10^3$ | $1.5 \times 10^2$ | $2.11 \times 10^5$ | $2.21 \times 10^5$ |
| Full-FT | $10 \times 10^3$ | - | - | $7.01 \times 10^5$ | $7.01 \times 10^5$ |

(*i.e., top-1 accuracy of 45% and 50%*) for *ImageNet*, **B2A@65** for *Caltech101*, and **B2A@90** for *CIFAR-10*, respectively. Figure 5 highlights the efficiency of VeCAF in comparison to other methods using such batch numbers. On ImageNet, VeCAF achieves 3.3× acceleration over full data finetuning (3075 *v.s.* 10250 batches) and outperforms other baselines with less batches as the target accuracy grows. Additionally, we plot the training loss as a function of the batch number for different approaches during the finetuning process in Figure 4. This visualization further highlights the faster convergence and performance of VeCAF when compared to other methods. These figures provide a comprehensive view of the performance and efficiency of VeCAF, emphasizing its enhanced efficiency in achieving the desired accuracy levels.

To verify that the amount of training batches serve as a good proxy for the overall computational cost of the finetuning process, we further analyze the overhead of using language encoder and performing data selection in each data selection loop in Table 4. We follow the descriptions in the PyTorch report [24] to estimate the computational cost of training and inference. The CLIP-ViT-L text encoder model we use in the experiments requires about 11× the FLOPs of the DeiT model being finetuned, but only needs to be inferenced once in each data selection loop. This leads to the FLOPs overhead of CEA to be approximately 4.5% of the total finetuning cost. The resulting reduction ratio of total FLOPs brought by VeCAF (7.01/2.21 = 3.17×) is therefore similar to the batch number reduction ratio (10/3 = 3.33×). Note that this analysis is consistent across datasets as all data are resized to 224×224 for ViT inference. This result verifies the computational efficiency of VeCAF as shown by the batch number reduction in previous experiments.

**Table 5: Top-1 accuracy of VeCAF with different types and sizes of PVM backbones, and choices of pretrained language encoders. ImageNet-1K results are reported with 1% data.**

| PVM \ LM | CLIP | BERT-L | mT5-L | GPT2-L |
|---|---|---|---|---|
| DeiT-S | $53.51_{\pm0.02}$ | $53.64_{\pm0.03}$ | $53.82_{\pm0.03}$ | $53.96_{\pm0.03}$ |
| DeiT-B | $58.31_{\pm0.04}$ | $58.63_{\pm0.04}$ | $58.71_{\pm0.03}$ | $58.87_{\pm0.01}$ |
| SwinT-S | $53.66_{\pm0.03}$ | $53.71_{\pm0.03}$ | $53.89_{\pm0.02}$ | $54.01_{\pm0.02}$ |
| SwinT-B | $56.76_{\pm0.01}$ | $56.87_{\pm0.03}$ | $56.98_{\pm0.03}$ | $57.13_{\pm0.02}$ |
| XciT-M | $58.48_{\pm0.04}$ | $58.70_{\pm0.03}$ | $58.77_{\pm0.03}$ | $58.95_{\pm0.03}$ |
| XciT-L | $61.13_{\pm0.01}$ | $61.37_{\pm0.01}$ | $61.56_{\pm0.03}$ | $61.78_{\pm0.03}$ |

**Table 6: Ablation study for the proposed techniques in VeCAF. Data selection ratio is set to 1% for CIFAR-10, 5% for Caltech101, and 1% for ImageNet-1K in each loop.**

| ODS | CEA | CIFAR-10 | Caltech-101 | ImageNet-1K |
|---|---|---|---|---|
| - | - | $92.79_{\pm0.12}$ | $60.55_{\pm0.10}$ | $55.67_{\pm0.13}$ |
| ✓ | - | $93.27_{\pm0.03}$ | $64.11_{\pm0.04}$ | $57.73_{\pm0.04}$ |
| - | ✓ | $93.15_{\pm0.05}$ | $63.04_{\pm0.06}$ | $56.13_{\pm0.05}$ |
| ✓ | ✓ | $\mathbf{93.57}_{\pm0.02}$ | $\mathbf{65.15}_{\pm0.03}$ | $\mathbf{58.31}_{\pm0.04}$ |

## 4.3 Results Analysis

*Generality of VeCAF.* The proposed VeCAF framework can be used to finetune various PVMs with the help of different language encoders (LMs). Table 5 reports results on ImageNet-1K using various pairs of PVMs (DeiT-S/B [38], Swin-Transformer-S/B [28], and XciT-M/L [2]) and LMs (BERT-L [9], mT5-L [42], and GPT2-L [34]). VeCAF demonstrates its ability to adapt to different PVM model architectures and the flexibility to be used with various text encoder models. These results prove VeCAF to be a general active data selection method for efficient PVM finetuning.

*Embedding visualization.* In Figure 6, we present the image embedding visualization of the CIFAR-10 training set using UMAP dimension reduction. With each method selecting 1% of data from the training set, black dots represent the samples selected by both ActiveFT and VeCAF, red stars denote samples only selected by VeCAF, and blue stars denote samples only selected by ActiveFT. While maintaining the diversity of data selection, samples chosen by VeCAF appear to be closer to the boundaries compared to those selected by ActiveFT. This confirms that the proposed ODS helps to select important samples around the decision boundaries. This is a result of our selection strategy that incorporates training objective and, therefore, helps the PVM to learn the subtle differences between categories more efficiently during the finetuning phase.

*Ablation study.* We first verify the importance of proposed ODS and CEA techniques in Table 6. Specifically, ODS can be disabled by removing the $\mathcal{L}$ term in Equation (7). ODS significantly improves classification accuracy, especially with limited data, and expedites

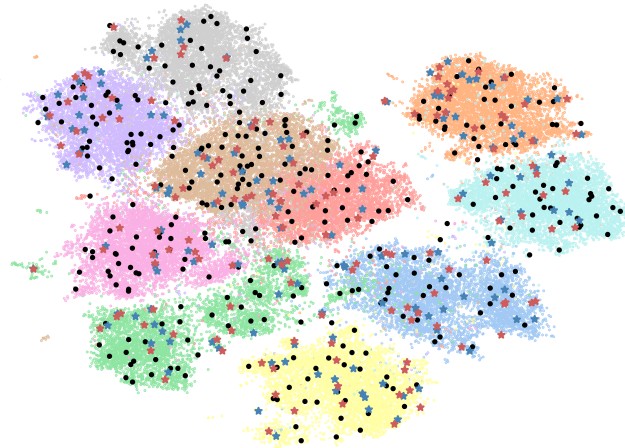

**Figure 6: UMAP visualization of training image embeddings. Each background color represents one class. For selected samples, ★ suggests being selected by VeCAF only, ★ suggests by ActiveFT [41] only, and • suggests by both.**

**Table 7: Ablation study of the number of data selection loops on the CIFAR-10 dataset with 5% data selection.**

| # Loops | 2 | 3 | 4 | 5 |
|---|---|---|---|---|
| CIFAR | $95.89_{\pm0.02}$ | $96.01_{\pm0.02}$ | $96.14_{\pm0.03}$ | $\mathbf{96.24}_{\pm0.03}$ |

the finetuning process. CEA further improves classification accuracy by integrating rich semantic information from text embeddings which leads to an enhanced model generalization by capturing underlying semantics in images.

Table 7 presents a study on the impact of the loop count in the multi-run selection setting using CIFAR-10 dataset. The total number of training batches is the same in all experiments with uniform division by the loop count. It is clear that the model performance is enhanced with the increase in the number of loops. However, performing additional loops of data selection leads to overhead in the total finetuning time. Therefore, we set the number of loops to three throughout our experiments to balance the trade-off between the model performance and the finetuning time.

## 5 CONCLUSIONS

In this paper, we proposed novel active data selection framework called VeCAF that improves the efficiency of model finetuning with two major components. First, VeCAF finds a subset of training data that leads to faster convergence using the objective-aware selection model. Second, it utilizes the text-domain knowledge in pretrained language encoders to augment image embeddings during selection.

Extensive experiments demonstrated advantages of the proposed approach such as more accurate in-distribution classification, robustness to out-of-distribution data and high computational efficiency. In future, we envision VeCAF can be extended to other multimedia domains with the potential of text-domain augmentation and by controllable active data generation for additional improvements.

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
