# OpenReview forum: "VeCAF: Vision-language Collaborative Active Finetuning with Training Objective Awareness"
_acmmm.org/ACMMM/2024/Conference — MM2024 Poster_

### Official Review · Reviewer_HVCd · 2024-05-23

**Rating:** 3
**Confidence:** 4

**Summary:**

This paper proposed a novel Vision-language Collaborative Active Finetuning (VeCAF) approach to efficiently select training subsets that are most significant to downstream finetuning objective. By introducing additional semantic information via pretrained language encoders to construct Cross-attentive Embedding Augmentation (CEA), VeCAF demonstrated superior performance and computational efficiency compared to baseline methods.

**Strengths:**

1.	Text-domain augmentation in image data selection.
2.	Efficient and effective end-to-end active fine-tuning framework.

**Limitations:**

1.	The concept of Objective-aware Data Selection (ODS) is like that of ActiveFT. It is recommended that the authors emphasize the difference between this part of the design and ActiveFT.
2.	The method for generating image captions for image classification datasets (CIFAR-10, Caltech101, and ImageNet-1K) is unclear. It is recommended that the authors add ablation experiments on the effect of image depiction quality on the performance of VeCAF.
3.	The generalizability experiment of the VeCAF method is incomplete. The effectiveness of VeCAF on other visual tasks (e.g., semantic segmentation) and backbone network structures (e.g., CNNs) is unknown.

**Suitability:**

2

---

### Official Review · Reviewer_jmPd · 2024-05-23

**Rating:** 4
**Confidence:** 1

**Summary:**

In this paper, the authors proposed novel active data selection framework called VeCAF that improves the efficiency of model finetuning with
two major components. First, VeCAF finds a subset of training data that leads to faster convergence using the objective-aware selection
model. Second, it utilizes the text-domain knowledge in pretrained language encoders to augment image embeddings during selection.
Extensive experiments demonstrated advantages of the proposed approach.

**Strengths:**

a）The experiment in this paper is thorough, and the presentation is relatively clear.

b)  On ImageNet, VeCAF uses up to 3.3× less training batches to reach the target performance compared to full finetuning, and achieves an accuracy improvement of 2.7% over the state-of-the-art active finetuning method with the same number of batches.

**Limitations:**

a) Please attach the publication time of the compared model  in the Table 1 for easy reading.

b)  The presentation of Figure 2 is not very clear compared to the corresponding textual expression,  and further polish is necessary.

**Suitability:**

3

---

### Official Review · Reviewer_uB78 · 2024-05-25

**Rating:** 4
**Confidence:** 3

**Summary:**

This paper proposes a VeCAF strategy to select significant data points and only uses these selected data for efficient downstream tuning. Extensive experiments show the leading performance and high computational efficiency of VeCAF.

**Strengths:**

1. The paper is well-written and easy to follow.
2. Using text embedding to augment the image embedding and improve out-of-distribution scenarios is interesting and insightful.
3. The work has conducted very thorough and comprehensive experiments, which is worthy of praise.

**Limitations:**

1. The details on how text embedding can help with data selection and fine-tuning are not clear enough. Based on the information provided in the paper (lines 399-403 and Algorithm 1), it seems that the text embedding is used to further enhance the already selected images, rather than to help with the initial data selection process. This makes me a little unclear about what it role is in the data selection process.
2. Introducing text embedding may raise some fairness issues. Specifically, the use of additional external information leads to unfairness in performance comparisons. Furthermore, I would like to know if this also incurs additional computational resource overhead.

**Suitability:**

2

---

### Meta-Review · Area_Chair_gBrz · 2024-07-01

**Recommendation:** Accept (Poster)
**Confidence:** 5

**Metareview:**

This paper ultimately received three positive "borderline accept" scores. After the rebuttal, all reviewers were satisfied with the authors' responses and the additional experimental results. Considering that the reviewers have no further questions, I recommend accepting this paper. I also suggest that the authors include the additional experiments in the final version of the paper.